# Synthesis of Fully Deacetylated Quaternized Chitosan with Enhanced Antimicrobial Activity and Low Cytotoxicity

**DOI:** 10.3390/antibiotics11111644

**Published:** 2022-11-17

**Authors:** Yeon Ho Kim, Ki Sun Yoon, Sung-Jae Lee, Eun-Jung Park, Jong-Whan Rhim

**Affiliations:** 1Department of Food and Nutrition, Kyung Hee University, 26 Kyungheedae-ro, Dongdaemun-gu, Seoul 02447, Republic of Korea; 2Department of Biology, Kyung Hee University, 26 Kyungheedae-ro, Dongdaemun-gu, Seoul 02447, Republic of Korea; 3Department of Biochemistry, School of Medicine, Kyung Hee University, 26 Kyungheedae-ro, Dongdaemun-gu, Seoul 02447, Republic of Korea

**Keywords:** chitosan, quaternization, deacetylation, antimicrobial activity, RNA expression, cytotoxicity

## Abstract

Fully deacetylated quaternary chitosan (DQCTS) was prepared by replacing the carboxyl group of chitosan with a quaternary ammonium salt. The DQCTS was characterized by Fourier transform infrared spectroscopy (FT-IR), X-ray diffraction (XRD), thermogravimetric analysis (TGA), and nuclear magnetic resonance (NMR). The antimicrobial activity of DQCTS was evaluated using the minimum inhibitory concentrations (MIC) methods and time-kill assay. DQCTS exhibited strong antibacterial and antifungal activity against *Staphylococcus aureus*, *Escherichia coli* O157: H7, *Candida albicans*, and *Aspergillus flavus*. Especially, the antifungal activity against *C. albicans* of DQCTS was greatly improved at 15.6 µg/mL of MIC and 31.3 µg/mL of minimum fungicidal concentration (MFC). Expression levels of virulence genes of microorganisms were also significantly decreased by DQCTS treatment, and the risk of virulence of microorganisms might be decreased. The result of the cytotoxic effect of DQCTS on human skin cells (HaCaT cells) indicated that the cytotoxicity of DQCTS on HaCaT cells was nearly non-toxic at 50 μg/mL. The DQCTS, with strong antimicrobial and low toxicity, has a high potential for use in functional food packaging and biomedical applications.

## 1. Introduction

Chitin, a linear high-molecular-weight polysaccharide composed mainly of (1-4)-linked 2-acetamido-2-deoxy-β-d-glucopyranose, is the second most abundant natural biopolymer after cellulose [1]. It is present in the exoskeletons of crustaceans (crabs and shrimps), insects, fungi and yeast cell walls [2]. Chitosan (CTS) is the deacetylated derivative of chitin, mainly composed of (1-4)-linked 2-amino-2-deoxy-β-d-glucopyranose [3,4]. The main structural difference between chitin and chitosan is the degree of acetylation (DA). In general, the DA of chitin is 90% or more, and the DA of chitosan is less than 40% [5]. The degree of deacetylation of chitosan determines the number of amino groups (NH_2_) present in the chitosan molecule, which is related to the biological activity of chitosan [6]. Protonation of amino groups by deacetylation increases the polyelectrolyte charge, resulting in property changes of chitosan, structure, and applications [7]. The multi-positively charged amino group at the C2 position of chitosan can interact with negatively charged compounds and macromolecules on the bacterial cell surface [1]. Therefore, chitosan with multiple positive charges has stronger antibacterial activity against Gram-negative bacteria than Gram-positive bacteria due to the difference in the outer membrane destruction [8]. In general, chitosan is classified into low, medium, high, and ultra-high deacetylated chitosan, depending on the degree of N-deacetylation of 55–70%, 70–85%, 85–95%, and 95–100% [7]. Fully deacetylated chitosan (DCTS) has the most potent biological activity and is expected to be readily chemically modified.

However, chitosan can only be dissolved under acidic conditions, and its antibacterial activity begins to decline above pH 6.5, limiting its use [9]. Therefore, various methods have been used to increase the solubility and functionality of chitosan. The quaternization of chitosan is interesting as one of the ways to increase the function of chitosan. Quaternized chitosan (QCTS), in which the quaternary ammonium groups substitute with the amine group in the chitosan, is water-soluble chitosan with excellent antibacterial and antifungal activity [10,11]. In addition, quaternized chitosan is known to have antibacterial activity in all pH ranges [12]. The kind and degree of substitution of quaternization affected water-soluble and antimicrobial activity [13]. The quaternization substituting trimethyl chitosan iodide results in higher water-soluble than quaternized by O-methylation [14,15]. Additionally, the degree of substitution by quaternarization is affected by the DA of chitosan, and the degree of quaternization of deacetylated chitosan is higher than chitosan without deacetylation. Therefore, the quaternization after fully deacetylated chitosan shows high water-solubility and antimicrobial activity compared with quaternized chitosan without full deacetylation. In order to increase the utility of chitosan, there is an increasing interest in the fabrication of DQCTS (fully deacetylated quaternary chitosan) and research on its antibacterial properties and the safety of its use.

Therefore, the main objective of this study was to prepare DQCTS to increase the functionality and water-solubility of chitosan. The antibacterial activity of the prepared DQCTS against *Staphylococcus aureus* and *E. coli* O157: H7 and the antifungal activity against *Aspergillus flavus* and *Candida albicans* were evaluated. The in vitro cytotoxic effects of skin cell cultures of DQCTS were also tested.

## 2. Results and Discussion

### 2.1. Characterization of Chitosan Compounds

The chemical structure of the chitosan compounds (CTS, QCTS, DCTS, and DQCTS) was evaluated using FT-IR, and the results are shown in Figure 1a. The peaks in the CTS and QCTS at 1661 and 1645 cm^−1^ were due to the C=O of an amide and the N-H binding of a primary amine, respectively [9]. The QCTS and DQCTS showed an absorption band at 1560 cm^−1^ due to the N-H deformation of the amino groups from the primary amine to a secondary amine [9]. The bands at 1061 and 1029 cm^−1^ were attributed to the symmetric C-O-C and C-O stretching, respectively [10,16]. The band at 1658 cm^−1^ corresponding to C=O disappeared, and the vibrational peak at 1597 cm^−1^ according to the free amino group increased, indicating that the structure of chitosan changed from CTS to DCTS [7]. However, the stretching vibration peak of DQCTS increased at 1658 cm^−1^ and decreased at 1597 cm^−1^ due to the quaternization of chitosan. Additionally, the DA values of chitosan were 78.71% for CTS and 100.00% for DCTS. These phenomena were the same as the previous work [7].

The powder X-ray diffraction (XRD) patterns of CTS, QCTS, DCTS, and DQCTS are shown in Figure 1b. The characteristic crystalline peaks of CTS and DCTS were shown at 2θ = 13° and 20°, respectively. The intensity at 13° decreased with the increase in the deacetylation of chitosan [17]. The degree of crystallinity of chitosan is a function of the degree of deacetylation as a semi-crystalline polymer [18]. However, both QCTS and DQCTS exhibited only one broad diffraction peak at 22°, indicating an amorphous structure. The crystallization capacity of CTS increased due to the intramolecular hydrogen bonding between the OH and NH_2_ groups [19]. On the other hand, quaternization broke intramolecular hydrogen bonds and reduced crystallinity [20]. The functional groups of chitosan, such as hydroxyl (OH) and amine (NH_2_) groups in the amorphous state, induce chitosan to easily react with other reagents or form complexes with various metal ions, making it more flexible and active [21].

Figure 1c shows the TGA and DTG thermograms of the CTS, QCTS, DCTS, and DQCTS. All chitosan compounds showed a two-step pattern of thermal degradations. The first weight change was observed at 80–100 °C due to moisture evaporation in the chitosan. Different patterns of main thermal degradation were observed between quaternized chitosan (QCTS and DQCTS) and non-quaternized chitosan (CTS and DCTS). The main thermal degradation was due to the decomposition of the polysaccharide chains, which occurred at 240 °C for QCTS and DQCTS and 300 °C for CTS and DCTS [22]. The thermal stability of chitosan decreased by the quaternization, probably due to the reduced crystallinity of the quaternized chitosan, as shown in the XRD results [20].

Figure 1d shows the ^1^H NMR spectra of CTS, QCTS, DCTS, and DQCTS. The characteristic signal of quaternization was the peak at 3.1 ppm (^+^N(CH_3_)^3^) [9]. Evidence of deacetylation of chitosan, such as DCTS, can be confirmed by a peak at 2.87 ppm corresponding to N-acetyl, which was removed by deacetylation of CTS [21]. The following signals confirm this: CH_3_ in acetamide (δ = 1.9 ppm), N-acetyl (δ = 2.87 ppm), and H^1^ of chitosan units (δ = 4.73 ppm) [10,21]. Additionally, chitosan’s degree of quaternization (DQ) values were 0%, 69.71%, 0%, and 99.15% of CTS, QCTS, DCTS, and DQCTS, respectively. The DQ value of DQCTS was higher than QCTS. These phenomena might cause more quaternary ammonium salts to be substituted to NH_2_ of DQCTS than QCTS.

### 2.2. Antimicrobial Activity

#### 2.2.1. Minimum Inhibitory Concentrations (MIC), Minimum Bactericidal Concentrations (MBC), and Minimum Fungicidal Concentrations (MFC)

Results of MIC, MBC, and MFC of the chitosan (CTS) and functionalized chitosans (QCTS, DCTS, and DQCTS) are shown in Table 1. The negative control did not show the antimicrobial activities. All chitosan compounds showed distinctive antimicrobial activities against test microorganisms except *A. flavus*. Among the chitosan compounds, CTS showed the lowest antimicrobial activities against *S. aureus*, *E. coli* O157: H7, and *C. albicans* showing the 750/1000 µg/mL for MIC/MBC of *S. aureus* and 1000/1000 µg/mL for MIC/MBC of *E. coli* O157:H7, and 1000/>1000 µg/mL for MFC of *C. albicans*. On the other hand, QCTS showed strong antibacterial activity against Gram-positive and Gram-negative bacteria. In addition, deacetylation of chitosan increased antifungal activity against *C. albicans*. DQCTS showed the strongest antimicrobial activities against *S. aureus*, *E. coli* O157: H7, and *C. albicans*. As can be seen from the MIC/MBC values of 250/250 µg/mL, there was no difference in the antibacterial activity of DQCTS against *S. aureus* and *E. coli*. However, the antifungal activity of DQCTS was stronger against *C. albicans* than bacteria, as shown by the MIC/MFC values of 15.6/31.3 µg/mL, possibly due to the difference in the microbial cell wall between bacteria and fungi [23]. The bacteria’s cell wall comprises peptidoglycan, an acidic polymer such as lipoteichoic acid, teichoic acid, and teichuronic acid. In contrast, the cell wall of fungi is composed of polysaccharides such as chitin, β-1,3-glucan, and mannose-containing glycoproteins [24]. Thus, bacteria are negatively charged due to the acidic polymers, and fungi are positively charged due to polysaccharides. However, *C. albicans* is negatively charged due to the colloid system in the cell surface [25]. Therefore, deacetylated chitosan showed stronger antibacterial activity because the structure of deacetylated chitosan changed from NH_2_ to NH^+^ and had a stronger positive charge than chitosan. The high antibacterial activity of QCTS is because the quaternized chitosan is easily absorbed and penetrates the microbial cell wall, resulting in leakage of proteins and intracellular components [23]. Sagoo et al. [26] also observed that the chitosan exhibited stronger antimicrobial activity against spoilage yeasts than bacteria.

#### 2.2.2. Time Kill Assay

The antimicrobial activity of the chitosan compounds was assessed using a total colony count method, and the results are shown in Figure 2. The negative control was not shown the antimicrobial activities. All chitosan compounds showed antimicrobial activity against all test microorganisms. The unmodified chitosan (CTS) inhibited only the growth of *S. aureus* and *E. coli* O157: H7. The functionalized chitosan (QCTS, DCTS, and DQCTS) showed stronger antimicrobial activity to stop the growth of *S. aureus* within 3 h than CTS. QCTS and DCTS reduced *E. coli* O157: H7 by 2 log CFU/mL, whereas DQCTS halted *E. coli* O157: H7 growth within 12 h of treatment. Against fungi, CTS reduced *C. albicans* by 1 log CFU/mL within 3 h, whereas QCTS, DCTS, and DQCTS completely stopped the growth of *C. albicans* within 12 h, 9 h, and 9 h, respectively. The non-deacetylated chitosan (CTS and QCTS) showed a slightly reduced growth rate of *A. flavus* compared with the control group. In contrast, DCTS and DQCTS showed a more distinctive growth reduction. DQCTS exhibited the strongest antifungal activity against *A. flavus* and stopped the growth of *A. flavus* after 2 days of incubation. These antimicrobial test results agree with the MIC, MBC, and MFC tests (Table 1).

#### 2.2.3. Effect of Fully Deacetylated Quaternized Chitosan (DQCTS) on the Morphology of Microorganisms

Morphological changes of microorganisms before and after treatment with DQCTS were observed using SEM (Figure 3). Normal cell membranes showed an intact surface, but all cell membranes were significantly altered after DQCTS treatment. By treatment with DQCTS, cocci bacteria, *S. aureus*, were contracted (Figure 3a,b), and rod cells, *E. coli* O157: H7, were also cracked on the surface (Figure 3c,d), showing the damage to the cell membrane of these bacteria by DQCTS. On the other hand, SEM results for fungi are different from bacteria. The surface of *C. albicans* was destroyed by DQCTS treatment, showing large holes (Figure 3e,f). Both spores and mycelia of *A. flavus* treated with DQCTS showed dissolved structure by destruction (Figure 3g,h), which explains the antifungal activity of DQCTS against *A. flavus* in the result of the time-kill assay (Figure 2).

The antibacterial action of DQCTS was confirmed by cell membrane permeation and degradation. Destruction, shrinkage, and cracking of microbial cell walls can cause death. The SEM results confirmed that *C. albicans* is more susceptible to chitosan compounds than bacteria. Although the exact antibacterial mechanism for chitosan has not yet been elucidated, it may explain several possibilities for how DQCTS acts on bacteria and fungi. First, the enhanced positive charge of the fully deacetylated chitosan may be strong ionic interaction with the charge of the cell membrane of microorganisms. Kim et al. reported similar SEM results, suggesting that the cause of chitosan-coated sulfur nanoparticles may be an ionic interaction of the positive charge of chitosan (NH^+^) with the microbial membrane [8]. Second, the quaternization of chitosan (N^+^(CH_3_)) can improve its combination with the microbial cell walls. Quaternized chitosan is an uptake enhancer between epithelial cells [27,28].

#### 2.2.4. Relative Expression Levels of Catalase, Virulence Factor, and Stress Response Sigma Factors

Quantitative RT-PCR was performed to determine expression levels of catalase, stress response sigma factors, and virulence factors of *S. aureus*, *E. coli* O157: H7, *C. albicans*, and *A. flavus* after treatment with chitosan (Figure 4). For *S. aureus*, the gene expression levels of both FemA (synthesis of peptidoglycan) and sigB (a general stress response sigma factor) were significantly decreased (*p* < 0.05) after treatment with all chitosan compounds. Especially, the trends of expression levels of staphylococcal enterotoxin A (SEA) and the production of enterotoxin were different between quaternized (QCTS and DQCTS) and non-quaternized chitosan (CTS and DCTS). The expression levels of SEA treated with CTS and DCTS significantly increased, while those treated with QCTS and DQCTS significantly reduced (*p* < 0.05).

For *E. coli*, expression levels of both ipfA (a virulence factor) and sigB (a general stress response sigma factor) were significantly decreased (*p* < 0.05) by all types of chitosan compounds. In particular, the expression level of the group treated with DQCTS was about one-tenth of the control, showing the lowest level among all chitosan compounds. However, the expression levels of sigW (membrane stress response sigma factor) treated with QCTS, DCTS, and DQCTS were significantly (*p* < 0.05) higher than that treated with chitosan.

Expression levels of both CHK1 (a membrane stress factor) and Hog1 (a general stress factor) treated with QCTS and DQCTS in *C. albicans* were significantly increased (*p* < 0.05) by about 7-fold and 8-fold, respectively. However, the expression levels of PamA (a biofilm generator factor) were significantly increased (*p* < 0.05), especially, QCTS and DQCTS induced a higher expression of PamA with about 158-fold and 22-fold, respectively, than DCTS.

The tested genes in *A. flavus* were factors in producing aflatoxins such as AflaC, AflaO, and AlfaR. The expression of AflaC was significantly increased (*p* < 0.05) by the treatment with all types of chitosan. However, the expression levels of both AflaO and AflaR were significantly decreased (*p* < 0.05) after treatment with DQCTS.

It is thought that the antibacterial activity of chitosan can be explained as a result of the gene expression level in bacteria. The sigB gene is a central regulator of general environmental stresses such as high and low temperatures or pH, high ethanol or osmolarity, and oxidative stress [29]. Although sigB in *S. aureus* and *E. coli* O157: H7 was not affected by treatment with chitosan, FemA, a gene for the sign of building membrane in *S. aureus*, and sigW, which is membrane stress sigma factor in *E. coli*, were affected by chitosan. Chitosan affects the destruction of the cell membrane without environmental stress. Therefore, these results may provide a basis for the interaction of chitosan with bacterial membranes by ionic charge and quaternized structure.

The trends of virulence gene expression were different between Gram-positive and Gram-negative bacteria. The QCTS significantly damaged the virulence gene of staphylococcal enterotoxin A (SEA), whereas the increase of SEA levels after CTS and DCTS treatment.

Zhao also reported that the *Staphylococcus aureus* enterotoxin I (SEI) gene significantly increased by chitosan treatment more than 2-fold [30]. On the other hand, ipfA expression in *E. coli* O157: H7 was significantly inhibited by all types of chitosan compounds. These results suggest that chitosan has a stronger virulence gene inhibitor in Gram-negative bacteria than in Gram-positive bacteria. The chitosan significantly reduced (*p* < 0.001) the expression level of α-hemolysin (hlyS), S fimbriae (sfa), and fimbrial adhesins (aer), which are virulent genes of more than half [31].

The gene expression levels of *C. albicans* were similar to those of bacteria. High-osmolarity glycerol response pathway (Hog1) gene response in *C. albicans* is activated to various stress conditions [32]. In addition, *Candida albicans* histidine kinase (CHK1) gene in *C. albicans* regulates mannan and glucan biosynthesis in the cell wall [33]. The quaternized chitosan triggered the expression of Hog1 and CHK1. These results suggest that some of the quaternization can bind to the cell wall of *C. albicans* and destroy the cell wall under various stresses. Therefore, the expression levels of Hog1 and CHK1 treated with QCTS and DQCTS were higher than others.

*A. flavus* can produce a variety of aflatoxins. The total aflatoxin regulatory limit for nut products is 20 ppb in Korea and the United States [34,35]. Most importantly, the risk of total aflatoxin should be decreased with antifungal agents. The AflaC gene is involved in the sub-source of aflatoxin B1 and aflatoxin G1 production, and the AflaO gene is implicated as the main source of production of all aflatoxin types [36]. The DQCTS significantly inhibited the expression levels of both AflaO and AflaR. These results suggest that DQCTS can inhibit the production of total aflatoxins. Li et al. [36] also reported that chitosan packaging film containing turmeric essential oil decreased AflaO and AflaR but increased the expression level of AflaC gene.

### 2.3. Cytotoxicity Test

MTT test was performed using HaCaT cells to determine the cytotoxicity of the chitosan compounds, and the results are shown in Figure 5. The cell viability decreased sharply at 200 µg/mL of all chitosan compounds. Cell viability was 98.2, 86.8, 92.4, and 67.4% of control levels treated with CTS, QCTS, DCTS, and DQCTS at 50 μg/mL, respectively. Cells treated with chitosan compounds were observed under a phase-contrast microscope (Figure 6). Images of cells treated with CTS and DCTS were not different from the control. However, treatment with QCTS and DQCTS partially disrupts the cell wall, allowing DQCTS particles to penetrate the cell wall.

Wang et al. reported that cell viability treated with quaternized chitosan in epithelial type II (AT2) cells showed a significant difference depending on the molecular weight of chitosan [37]. They found that quaternized chitosan (QCTS) with a longer carbon chain (C18) showed a stronger cytotoxic effect than quaternized chitosan (QCTS) with a shorter carbon chain (C12). The cytotoxicity of the quaternized chitosan (QCTS and DQCTS) can be due to the permeation of chitosan through the cell wall, as shown in the phase-contrast microscopic images (Figure 6). The penetrated chitosan may impair the cell surfaces by cationic polymer aggregation [38]. These cytotoxicity tests agree with the antimicrobial activity of the chitosan compounds.

## 3. Materials and Methods

### 3.1. Materials

Chitosan from shrimp shells (75–85% degree of deacetylation and 1747.5 g/mol of molecular weight), 3-chloro-2-hydroxypropyl trimethylammonium chloride (CHPTAC) solution, and methylthiazolyldiphenyl-tetrazolium bromide (MTT) solution were obtained from Sigma-Aldrich (St. Louis, MO, USA). Acetic acid, sodium carbonate, methyl alcohol, sodium hydroxide, and iodine were purchased from Duksan Pure Chemicals Co., Ltd. (Ansan, Gyeonggi-do, Korea).

Tryptic soy broth (TSB), tryptic soy agar (TSA), yeast malt broth (YMB), and yeast malt agar (YMA) were purchased from KisanBio Co., Ltd. (Seoul, Korea). Potato dextrose broth (PDB), potato dextrose agar (PDA), and peptone water were gained from Difco (Becton Dickinson, Sparks, MD, USA). YM 3M Petri film was purchased from 3M Corporation (St. Paul, MN, USA). Test microbial strains, *Staphylococcus aureus* (ATCC 13565, enterotoxin A*), Escherichia coli* O157: H7 (ATCC 11234), *Candida albicans* (ATCC 18804), and *Aspergillus flavus* (ATCC 22546), were obtained from the Korean Culture Center of Microorganisms (KCCM, Seoul, Korea). Human keratinocyte HaCaT cells were provided by Dr S.J. Kim (CHA University, Seoul, Korea). Dulbecco’s modified Eagle’s medium (DMEM), fetal bovine serum (FBS), and 1% penicillin/streptomycin were purchased from Gibco (ThermoFisher Scientific, Waltham, MA, USA).

### 3.2. Preparation of Fully Deacetylated Chitosan

Fully deacetylated chitosan (DCTS) was prepared following He et al. [7]. Ten grams of chitosan were dispersed in 200 mL of 1 N of NaOH solution with stirring for 5 min. After the chitosan was completely dispersed, it was heated at 120 °C for 120 min using an autoclave (VS-1321-45 Series, Vision, Seoul, Korea). Then, the precipitate was centrifuged, washed with distilled water until neutral pH, filtered, and dried in an oven at 105 °C to get DCTS powder. To determine the DA of chitosan, the following formulation of Butchosa et al. [1] was used for calculation.
(1)$$\mathrm{DA}\;\mathrm{of}\;\mathrm{chitosan}=\{1-\left(\frac{A_{1645}}{A_{3287}}\times \frac{1}{1.33}\right)\}\times 100$$
where A_1645_ and A_3287_ are the absorbance of each sample using FT-IR at 1645 cm^−1^ and 3287 cm^−1^, respectively.

### 3.3. Preparation of Quaternized Chitosan

Quaternized chitosan (QCTS) and fully deacetylated quaternized chitosan (DQCTS) were prepared following the method of Sajomsang et al. [10] with slight modification. One gram of CTS or DCTS was dissolved in 50 mL of 1% acetic acid solution. The chitosan solution was added dropwise into 50% methanol solution containing 2% Na_2_CO_3,_ and collected the regenerated chitosan by filtration. Forty mL of CHPTAC solution was added to the reaction bottle, and the pH was adjusted to 8 using 20% NaOH solution. Then, the regenerated chitosan and 0.25 g of iodine were added to the reaction bottle and stirred at room temperature for 48 h, and added 50 mL of distilled water with stirring at 60 °C for 24 h. The solution was dialyzed against distilled water until neutral pH. The dialyzed solution was evaporated to dryness at 60°C using a rotary evaporator (N-1200A, Eyela, Shanghai, China). The concentrated solution was precipitated in acetone and dried in an oven at 50 °C for 3 days to get QCTS and DQCTS powder. The degree of quaternization (DQ) of chitosan was calculated using the following equation:(2)$$\mathrm{DQ}\;\mathrm{of}\;\mathrm{chitosan}=\{1-\left(\;\frac{\mathrm{NHAc}}{3\times b}\right)\}\times 100$$
where b and NHAc are the integral area at δ 4.2 ppm and δ 1.9 ppm of each sample using NMR, respectively.

### 3.4. Characterization of Functionalized Chitosan

Fourier transform infrared (FT-IR) spectra of chitosan (CTS) and functionalized chitosan (QCTS, DCTS, and DQCTS) were recorded using an FT-IR spectrometer (TENSOR 37 spectrophotometer with OPUS 6.0 software, Billerica, MA, USA) with an average resolution of 32 scans at 4 cm^−1^ from 4000 to 400 cm^−1^.

X-ray diffraction (XRD) patterns of the CTS, QCTS, DCTS, and DQCTS were analyzed using an X-ray diffractometer (PANalytical X’pert pro-MRD diffractometer, Amsterdam, Netherlands). The sample was dropped on a glass slide and air-dried, and the spectrum was obtained using Cu Kα radiation (wavelength of 0.1541 nm) equipped with a nickel monochromator operated at 40 kV and 30 mA. The diffraction pattern was recorded at a scanning speed of 0.4°/min with diffraction angles of 2θ = 30–80°.

For the thermal stability analysis of the chitosan (CTS, QCTS, DCTS, and DQCTS), 10 mg of each sample was added to a standard aluminum pan and heated from 30 °C to 600 °C at a heating rate of 10 °C /min under a nitrogen flow of 50 cm^3^/min using a thermogravimetric analyzer (Hi-Res TGA 2950, TA Instrument, New Castle, DE, USA). The derivative of TGA (DTG) was calculated using a central finite difference method, and the char content of the samples at 600 °C was determined from the TGA curve [39].

^1^H NMR spectra were analyzed using a 400 MHz FT-NMR spectrometer (AVANCE III HD 400, Bruker Biospin, MA, USA). All measurements were performed at 300 K using a pulse accumulating 64 scans with an LB parameter of 0.30 Hz. A five mg sample was dissolved in 1% (*v*/*v*) CD_3_COOD in D_2_O. ^1^H NMR spectra were used to determine the degree of quaternization of CTS and DCTS as the ratio between the peak integration of protons at 3.1 ppm from quaternary ammonium groups [40].

### 3.5. Preparation of Microbial Cultures and Assay Media

The microbial cultures were prepared following the method of Kim et al. [41]. The test bacteria (*Staphylococcus aureus* and *E. coli* O157: H7) were activated in TSB with 20% glycerol and *Candida albicans* in YM medium with 30% glycerol and stored in a −80 °C freezer. For antimicrobial testing, 10 µL of stock culture was inoculated with 10 mL of sterile medium (TSB for *S. aureus* and *E. coli* O157: H7, YMB for *C. albicans*) and incubated at 140 rpm for 24 h at 36 °C using a rotary shaker (VS-8480SR, Vision, Seoul, Korea). One mL of culture medium was added to 9 mL of 0.1% sterile peptone water and serially diluted to a 5–6 log CFU/mL for final concentration. This solution was used for antimicrobial activity tests.

*Aspergillus flavus* was kept at −20 °C in sterile potato dextrose broth (PDB) containing 50% glycerol solution. A hundred μL of the culture broth was inoculated on a potato dextrose agar (PDA) plate and incubated for 4 days in an incubator (VS-1203P4S, Vision, Seoul, Korea) at 25 °C. Then, 10 mL of 0.1% sterilized peptone water was added to *A. flavus* in a PDA plate. Spores of *A. flavus* were scraped using a sterile platinum loop, and 1 mL of solution was inoculated to a YM 3M Petri film to check the number of spores. One mL of the culture was diluted with 9 mL of 0.1% sterile peptone water to a final concentration of 4–5 log CFU/mL. This solution was used for antimicrobial activity tests.

### 3.6. Antimicrobial Activity

The antimicrobial activities of chitosan (CTS) and functionalized chitosan (QCTS, DCTS, and DQCTS) were evaluated using minimal inhibitory concentration, total viable colony count, and morphological methods. For this experiment, 15.6, 31.3, 62.5, 125, 250, 500, 750, and 1000 µL of the chitosan solutions were prepared, where CTS and DCTS were dissolved in 0.1% acetic acid solution, and QCTS and DQCTS were dissolved in distilled water. 0.1% acetic acid was used for negative control.

#### 3.6.1. Minimum inhibitory Concentrations (MIC), Minimum Bactericidal Concentrations (MBC), and Minimum Fungicidal Concentrations (MFC)

The MIC, MBC, and MFC of the chitosan solutions were determined using a broth microdilution method [42]. A fifty μL of each culture (5–6 log CFU/mL) of *S. aureus*, *E. coli* O157: H7, and *C. albicans* were transferred into 96 plate wells containing 100 μL of each medium broth with 50 μL of the chitosan solutions to make the final concentrations of 15.6, 31.3, 62.5, 125, 250, 500, 750, and 1000 µg/mL. The 96 plate wells were then incubated at 36 °C and 140 rpm for 24 h.

Optical density (OD) at 600 nm was measured using Multiskan GO (Thermo Scientific, MA, USA). The MIC was determined as the lowest concentration of the chitosan compounds that did not increase the OD value of the sample compared to the control. The MBC and MFC were the lowest concentration of chitosan compounds without any colony in each medium agar plate. The MIC and MFC of *A. flavus* were determined using the same procedure after incubation at 25 °C for 4 days.

#### 3.6.2. Time Kill Assay

Time kill assay of chitosan compounds was performed against *S. aureus*, *E. coli* O157: H7, *C. albicans*, and *A. flavus* following the method of Kim et al. [8]. For this, 2 mL of each culture (5–6 log CFU/mL) of *S. aureus*, *E. coli*, and *C. albicans* were inoculated into 16 mL of each broth containing 2 mL of chitosan compounds to make the final concentrations of 15.6, 31.3, 62.5, 125, 250, 500, 750, and 1000 µg/mL, and incubated at 36 °C and 140 rpm for 24 h. Samples were taken at predetermined time intervals (0, 3, 6, 9, and 12 h), diluted by serial dilutions, spread on each agar plate, incubated at 36 °C for 24 h, and viable colonies were counted. In the case of *A. flavus*, the same procedure was performed by culturing at 25 °C for 4 days. For comparison, medium broth without the chitosan compounds was tested following the same procedure.

#### 3.6.3. Effect of Chitosan Compounds on the Morphology of Microbial Cells

The antimicrobial action of DQCTS was investigated through the morphological change of microorganisms [43]. For this experiment, test microorganisms (~6 log CFU/mL) were incubated on each agar medium plate at 36 °C for 24 h (for *A. flavus* at 25 °C for 4 days), then treated with 2 mL of DQCTS to make a concentration of 250 µg/mL. Then, the agar medium was cut into 3 cm × 3 cm and fixed in 5% glutaraldehyde at room temperature for 2 h. The agar medium was washed with 0.1% peptone water, incubated in 1% osmium tetroxide for 1 h, and sealed with parafilm. After that, osmium tetroxide was removed using a pipette, and the agar medium was washed twice with distilled water for 20 min each. The agar medium was sequentially dehydrated for 30 min using 35, 50, 75, 95% (2 times), and 100% (3 times) ethanol. Then, hexamethyldisilazane was treated twice for 30 min each, and the agar medium was dried in a desiccator for 3 days. After that, it was mounted to the SEM stub with carbon tape and sputter-coated with platinum. The microbial cells’ morphology was then observed using a field emission scanning electron microscope (FE-SEM, Quanta 400 FEI, Hillsboro, OR, USA).

#### 3.6.4. Quantitative Real-Time PCR (qRT-PCR)

Cultured cells on agar plates of *S. aureus*, *E. coli* O157: H7, *C. albicans*, and *A. flavus* were treated with each chitosan compound at a 250 μg/mL concentration for 1 h. As a control group, an untreated sample of chitosan was used. The chitosan-treated samples were resuspended in a 10 mL phosphate-buffered solution (pH 7.5) containing chitosan of equal concentration. Total RNAs of *E. coli* and *S. aureus* were isolated using RiboEX reagent (GeneAll Biotechnology, Seoul, Korea) and purified using an RNeasy mini purification kit (Qiagen, Hilden, Germany). In the case of *C. albicans* and *A. flavus*, total RNA was extracted using only the RiboEX reagent. The quality and quantity of total RNA were confirmed using DS-11 Spectrophotometer (Denovix, Wilmington, DE, USA). For qRT-PCR analysis, cDNA was synthesized from 1 µg of purified total RNA using the PrimeScript 1st Strand cDNA Synthesis kit (TaKaRa, Shiga, Japan). The qRT-PCR analysis was gauged using TB Green Premix Ex Taq polymerase (TaKaRa, Japan) on Bio-Rad CFX96TM Optics Module (Bio-Rad, Hercules, CA, USA). The relative level of expression for each gene was calculated using the comparative threshold cycle method after normalizing the expression level of the gene encoding glyceraldehyde-3-phosphate dehydrogenase (GAPDH) for *E. coli* O157:H7 and *S. aureus* and an 18S ribosomal RNA for *C. albicans* and *A. flavus*, which maintains a stable level of expression at all stages of growth [44,45]. The virulence factors of *E. coli*, *S. aureus*, *C. albicans*, and *A. flavus* were chosen for long polar fimbriae A (IpfA), Staphylococcal enterotoxin A (SEA), plasma membrane ATPase pump (PamA), and aflatoxin biosynthesis (AflaC, AflaO, AflaR), respectively [46,47,48,49,50]. Furthermore, the stress response genes, including cell wall-related gene and sigma factor of bacteria, were chosen, and measured the relative expression levels during each chitosan nanoparticle treatment [29,51,52,53].

### 3.7. Cytotoxicity Test

#### 3.7.1. Cell Culture

HaCaT, a human keratinocyte cell line, cells were maintained in Dulbeco’s Modified Eagle Medium (DMEM) with 10% Fatal Bovine Serum (FBS), streptomycin (100 mg/mL), l-glutamine (4.5 mg/mL), and penicillin (100 units/mL) at 37 °C in a 5% CO_2_/95% air humidified atmosphere.

#### 3.7.2. Cell Viability

Cell viability of chitosan compounds was evaluated by MTT assay [54]. A 150 µL of cells (5 × 10^5^ cells/mL) was seeded into a 96-well plate and stabilized for 24 h. Cells were incubated in culture media (DMEM with 10% FBS) with 50 µL of each chitosan compound (the final concentrations of 0, 50, 100, and 200 µg/mL) for 24 h. The MTT solution (4 mg/mL, 20 μL/well) was added to each well and incubated at 37 °C for 4 h in the dark. The supernatants were aspirated, and dimethyl sulfide (150 µL/well) was added to solubilize formazan produced in the cells. The absorbance at 540 nm was measured using a multi-mode microplate reader (BioTek, Winooski, VT, USA). The viability of the treatment group (%) was calculated as a relative value based on the absorbance value of the control group.

### 3.8. Statistical Analysis

Antimicrobial tests were performed in triplicate, and the results were presented as mean ± SD (standard deviation). One-way analysis of variance (ANOVA) was performed, and the significance of each mean value was determined (*p* < 0.05) by Duncan’s multiple range tests using the SPSS statistical analysis computer program (SPSS Inc., Chicago, IL, USA).

## 4. Conclusions

Functionalized chitosan compounds (QCTS, DCTS, and DQCTS) were prepared using deacetylation and quaternization and confirmed through FT-IR, XRD, and NMR analysis. The biological and functional properties, such as antimicrobial activity and water solubility of chitosan, were significantly increased through the quaternization and deacetylation of chitosan. The fully deacetylated and quaternized chitosan (DQCTS) showed the strongest antibacterial and antifungal activity against *S. aureus*, *E. coli* O157: H7, *C. albicans*, and *A. flavus*. Especially, DQCTS showed the strongest antifungal activity against *C. albicans* at the MIC/MFC values of 15.6/31.3 µg/mL. The quaternized chitosan (QCTS and DQCTS) decreased the virulence gene expression of bacteria such as SEA and ipfA, and decreased the risk of total aflatoxins production by lowering the expression levels of both AflaO and AflaR. DQCTS exhibited low cytotoxicity against HaCaT cells at 50 μg/mL. This study suggests the benefits of DQCTS as antibiotics with water-solubility and broad and strong antimicrobial activity against bacteria and yeast. Moreover, fully deacetylated and quaternized chitosan is expected to have high potential in functional food packaging and biomedical applications.

## Figures and Tables

**Figure 1 antibiotics-11-01644-f001:**
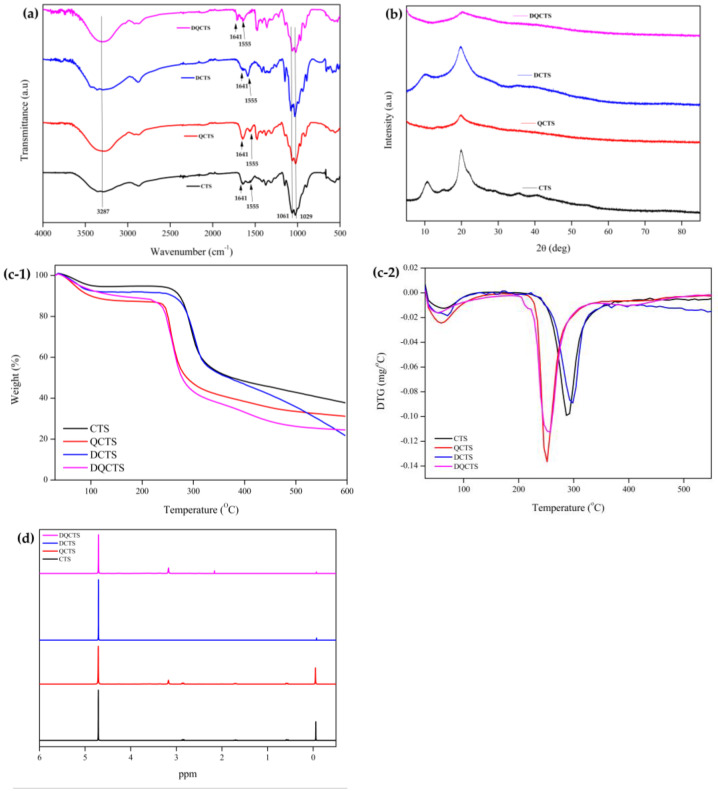
(**a**) FT-IR, (**b**) XRD pattern, (**c**) TGA (**c**-**1**) and DTG (**c-2**) thermograms, and (**d**) NMR of chitosan compounds. CTS: chitosan, QCTS: quaternized chitosan, DCTS: fully deacetylated chitosan, DQCTS: fully deacetylated quaternized chitosan.

**Figure 2 antibiotics-11-01644-f002:**
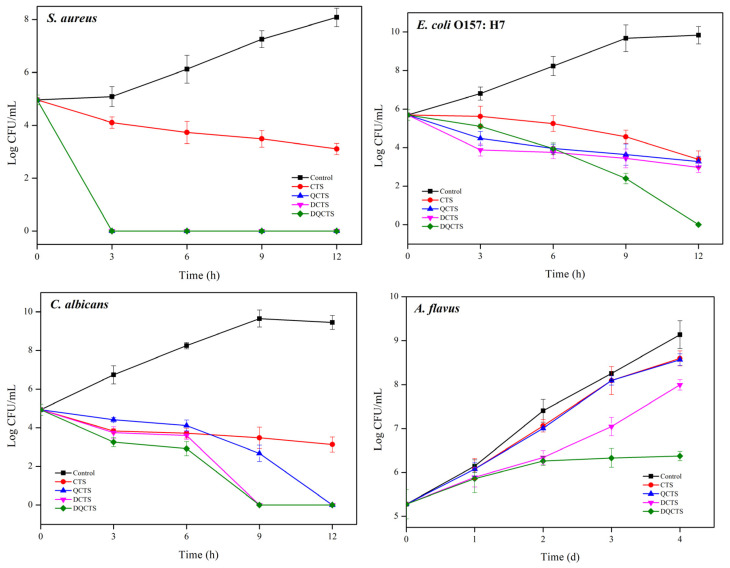
Antimicrobial activity of chitosan compounds against *S. aureus* (ATCC 13565), *E. coli* O157:H7 (ATCC 11234), *C. albicans* (ATCC 18804), and *A. flavus* (ATCC 22546). CTS: chitosan, QCTS: quaternized chitosan, DCTS: fully deacetylated chitosan, DQCTS: fully deacetylated quaternized chitosan.

**Figure 3 antibiotics-11-01644-f003:**
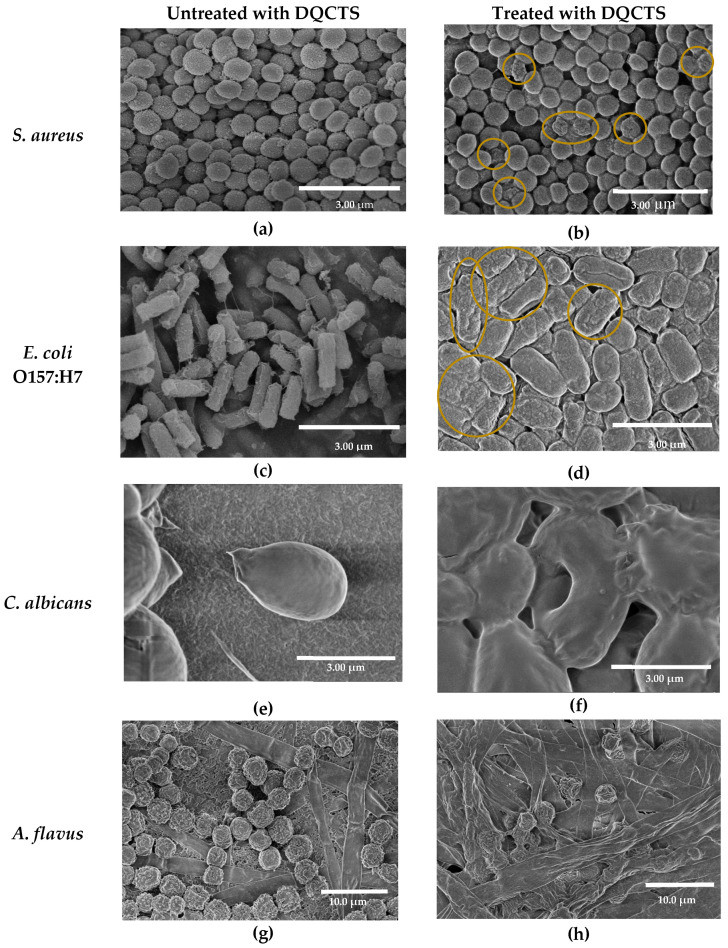
FE-SEM images of microorganisms before (**a**,**c**,**e**,**g**) and after treatment (**b**,**d**,**f**,**h**) with fully deacetylated quaternized chitosan (DQCTS).

**Figure 4 antibiotics-11-01644-f004:**
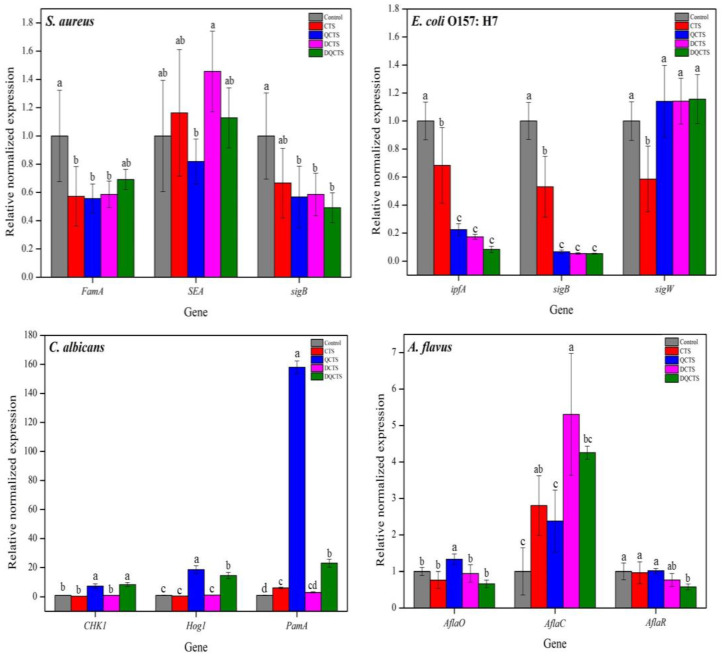
Comparison of relative expression levels of each gene treated with chitosan compounds using qRT-PCR in *S. aureus*, *E. coli* O157: H7, *C. albicans*, and *A. flavus*. Each bar represents relative expression levels compared to the GAPDH values (gray color, control). The values were presents as mean ± standard deviation (*n* = 3). a–d means in each bar with a different letter is significantly different by Duncan’s multiple range test at *p* < 0.05. CTS: chitosan, QCTS: quaternized chitosan, DCTS: fully deacetylated chitosan, DQCTS: fully deacetylated quaternized chitosan.

**Figure 5 antibiotics-11-01644-f005:**
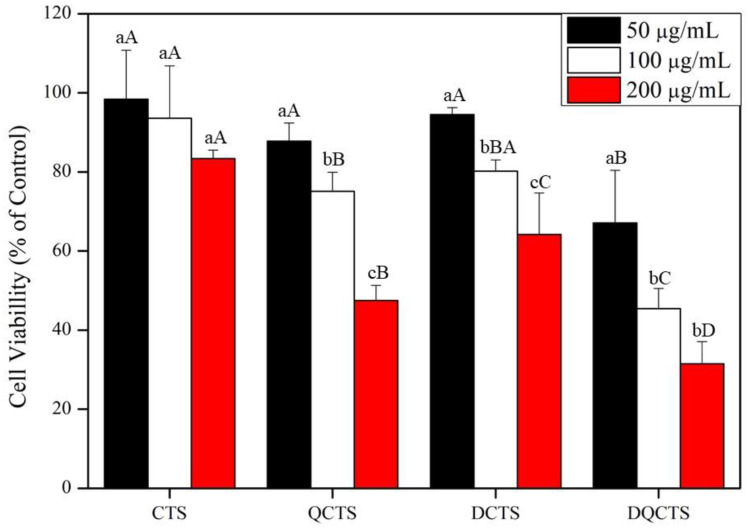
Cell viability of HaCaT cells treated with chitosan compounds at 50, 100, and 200 μg/mL. The results were presented as three independent experiments’ mean (*n* = 3) ± standard deviation (SD). a–c means values in each concentration with a different letter are significantly different by Duncan’s multiple range test at *p* < 0.05. A–D means values in each chitosan compound with a different letter are significantly different by Duncan’s multiple range test at *p* < 0.05. CTS: chitosan, QCTS: quaternized chitosan, DCTS: fully deacetylated chitosan, DQCTS: fully deacetylated quaternized chitosan.

**Figure 6 antibiotics-11-01644-f006:**
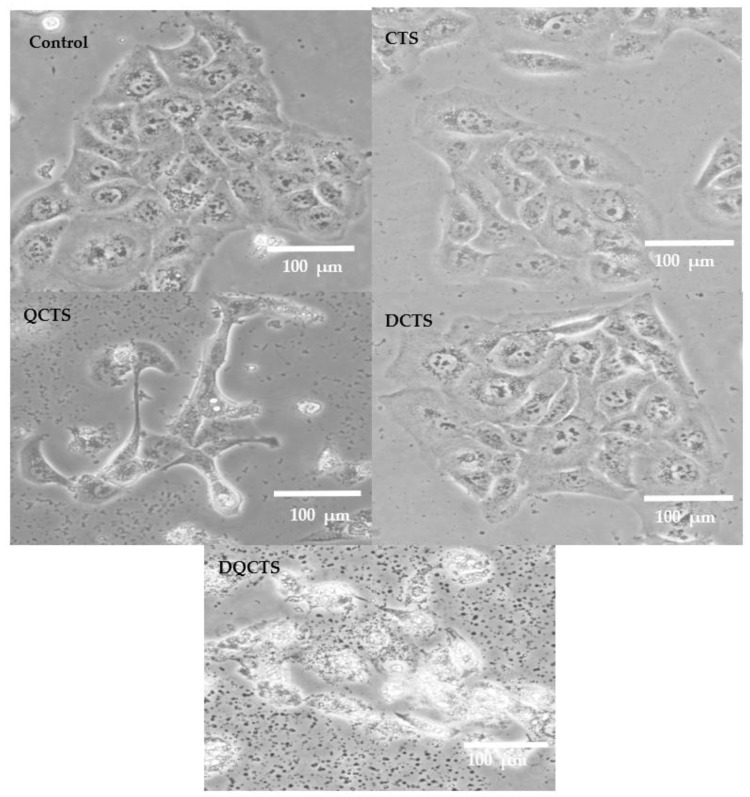
Phase-contrast microscopic images of HaCaT cells before and after treatment with chitosan compounds. CTS: chitosan, QCTS: quaternized chitosan, DCTS: fully deacetylated chitosan, DQCTS: fully deacetylated quaternized chitosan.

**Table 1 antibiotics-11-01644-t001:** Minimum inhibitory concentration (MIC), minimum bactericidal concentration (MBC), and minimum fungicidal concentration (MFC) of various types of chitosan compounds.

Microorganism	MIC/MB(F)C (μg/mL)			
	CTS	QCTS	DCTS	DQCTS
*S. aureus*	750 ^1^/1000 ^2^	250/250	750/750	250/250
*E. coli* O157:H7	1000/>1000	250/500	750/>1000	250/250
*A. flavus*	>1000/>1000 ^3^	>1000/>1000	>1000/>1000	>1000/>1000
*C. albicans*	1000/>1000	750/1000	62.5/125	15.6/31.3

^1^ Minimum inhibition concentration (µg/mL); ^2^ Minimum bactericidal concentration (µg/mL); ^3^ Minimum fungicidal concentration (µg/mL); CTS: chitosan, QCTS: quaternized chitosan, DCTS: fully deacetylated chitosan, DQCTS: fully deacetylated quaternized chitosan.

## Data Availability

All available data are reported in the manuscript.

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
