# Peer review of "Synthesis of Fully Deacetylated Quaternized Chitosan with Enhanced Antimicrobial Activity and Low Cytotoxicity"

_antibiotics, 2022, doi:10.3390/antibiotics11111644_

Round 1
Reviewer 1 Report
Abstract
Better provide numerical value of results and tested concentrations for antimicrobial and cytotoxicity assays.
Introduction
Add more information on the advantages of quarternary chitosan. How DQCT is superior over normal chitosan?
Results
For any antimicrobial activity, it is better to use standard antibiotics to test the comparative efficacy of our prepared particles. So that the antimicrobial effectiveness of prepared particles could be claimed. Why not the present study have used any standard antibiotics?
Conclusion
Rewrite this part. Provide a strong conclusion on how the prepared particles would be economical and more effective than standard drugs. How this study would benefit the society in the future? Add these information.
Author Response
Reviewer 1 Comments
- Abstract: Better provide the numerical value of results and tested concentrations for antimicrobial and cytotoxicity assays.
Response: We revised the abstract, including the numerical values of the results as recommended (Lines 13-20).
- Introduction: Add more information on the advantages of quaternary chitosan. How DQCT is superior to normal chitosan?
Response: We added more information on the advantages of quaterary chitosan (LL 53-63).
- Results: For any antimicrobial activity, it is better to use standard antibiotics to test the comparative efficacy of our prepared particles. So that the antimicrobial effectiveness of prepared particles could be claimed. Why not the present study has used any standard antibiotics?
Response: Yes, we agree that standard antibiotics for comparison can be used to test the antimicrobial activity of newly developed material. However, the antimicrobial activity of chitosan is well known, and chitosan itself can be considered a standard antimicrobial material like antibiotics. In this work, we focused on increasing the application of chitosan by deacetylation and quaternization. So we tested the antimicrobial activity after the quaterization of chitosan without comparing it with antibiotics.
- Conclusion: Rewrite this part. Provide a strong conclusion on how the prepared particles would be economical and more effective than standard drugs. How this study would benefit society in the future? Add this information.
Response: We revised the conclusions as recommended (LL. 520-533).
Reviewer 2 Report
This manuscript prepared fully deacetylated quaternized chitosan with high antimicrobial activity and low cytotoxicity, the experiments were well designed. But following issue need to be addressed well.
1. The unit format of “cm-1”, “NH2”and “NH+” are wrong, please go through the whole manuscript and revise them accordingly.
2. How do you know the chitosan was fully deacetylated quaternized based on the data shown in Fig 1? Please explain.
3. Fig 4 and Fig5, please indicate what’s “a, ab, b, c, etc” refers to in the caption.
4. Please include scale bar in Fig6.
5. What’s the molecular weight of prepared different types of chitosan?
Author Response
Reviewer 2 Comments
This manuscript prepared fully deacetylated quaternized chitosan with high antimicrobial activity and low cytotoxicity. The experiments were well-designed. But the following issue needs to be addressed well.
- The unit format of “cm-1”, “NH2”and “NH+” are wrong, please go through the whole manuscript and revise them accordingly.
Response: We correcetd the unit format in the whole manuscript.
- How do you know the chitosan was fully deacetylated quaternized based on the data shown in Fig 1? Please explain.
Response: We appreciate the reviewer’s insightful comment. We used the method from the following paper (http://dx.doi.org/10.1016/j.ejar.2015.09.003, reference 1), which provides the formulation for the degree of acetylation (DA) of chitosan.
We calculated the DA of chitosan according to the formulation and confirmed the 100% deacetylation of chitosan. We added this information to the revised manuscript (LL. 83-85 and 349-355).
Also, we calculated the degree of quaternization (DQ) using a formulation from the following paper (https://doi.org/10.1016/j.ijbiomac.2009.03.003, reference 2) (LL. 112-116 and 369-375).
- Fig 4 and Fig 5, please indicate what’s “a, ab, b, c, etc” refers to in the caption.
Response: They indicate statistical differences between treatment groups. We added this information in the caption of Fig. 4 and Fig. 5.
- Please include the scale bar in Fig. 6.
Response: Added the scale bar in Fig. 6.
- What’s the molecular weight of prepared different types of chitosan?
Response: We added the molecular weight information (LL. 328-329).
Round 2
Reviewer 1 Report
The authors have revised the manuscript well by addressing the reviewer's comments. All the comments were satisfactorily incorporated and modified accordingly.
Author Response
The authors have revised the manuscript well by addressing the reviewer's comments. All the comments were satisfactorily incorporated and modified accordingly.
Response: We appriciate your comments. We corrected minor parts such as spell check.